# Training Simultaneous Speech Translation
# with Robust and Random Wait-$k$-Tokens Strategy

**Linlin Zhang**[1*], **Kai Fan**[2*†], **Jiajun Bu**[1], **Zhongqiang Huang**[2]

[1]Zhejiang University, [2]Alibaba DAMO Academy

{zhanglinlinlin,bjj}@zju.edu.cn

{z.huang,k.fan}@alibaba-inc.com

## Abstract

Simultaneous Speech Translation (SimulST) is a task focused on ensuring high-quality translation of speech in low-latency situations. Despite this, the modality gap (*e.g.*, unknown word boundaries) between audio and text presents a challenge. This gap hinders the effective application of policies from simultaneous text translation (SimulMT) and compromises the performance of offline speech translation. To address this issue, we first leverage the Montreal Forced Aligner (MFA) and utilize audio transcription pairs in pre-training the acoustic encoder, and introduce a token-level cross-modal alignment that allows the wait-$k$ policy from SimulMT to better adapt to SimulST. This token-level boundary alignment simplifies the decision-making process for predicting read/write actions, as if the decoder were directly processing text tokens. Subsequently, to optimize the SimulST task, we propose a robust and random wait-$k$-tokens strategy. This strategy allows a single model to meet various latency requirements and minimizes error accumulation of boundary alignment during inference. Our experiments on the MuST-C dataset show that our method achieves a better trade-off between translation quality and latency.

## 1 Introduction

Simultaneous Speech Translation (SimulST) is a task designed to generate real-time translations by incrementally consuming audio frames. A common practice is to cascade a streaming Automatic Speech Recognition (ASR) system and a Simultaneous Text Machine Translation (SimulMT) model (Oda et al., 2014; Dalvi et al., 2018). The latter has been significantly improved earlier, as evidenced by (Gu et al., 2017). Then, the prefix-to-prefix (P2P) framework (such as the wait-$k$ policy) (Ma et al., 2019a; Le et al., 2020) has been developed to

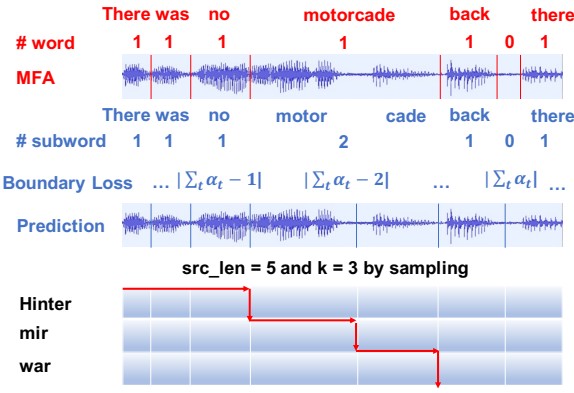

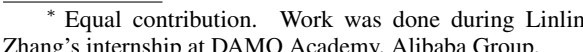

Figure 1: An example of our proposed robust and random wait-$k$-tokens strategy. #word $= 0$ indicates that MFA aligns a chunk of consecutive frames to no word. $\alpha_t$ is the CIF weight (details refer to Sec. 3.1).

reduce the discrepancy between training and inference. An efficient wait-$k$ method was proposed by (Elbayad et al., 2020) to train multiple paths by randomly sampling $k$ values, instead of training multiple models *w.r.t.* different $k$ values. The progress in SimulMT has greatly influenced advancements in SimulST. In particular, transformer-based end-to-end neural architectures have achieved performance levels comparable to cascade systems in both offline and simultaneous ST tasks (Vila et al., 2018; Bentivogli et al., 2021; Fang et al., 2022a; Ren et al., 2020; Liu et al., 2021; Anastasopoulos et al., 2022).

Typically, an end-to-end SimulST model requires a high-quality speech translation model that can take as input partial audio and a decision policy for controlling read/write actions. One simple but effective approach is to borrow ideas from SimulMT. However, applying SimulMT strategies directly to SimulST poses a challenge because speech signals are continuous, while text tokens are discrete and have inherent meanings. Therefore, the strategies of SimulMT, which defines input granularity at the token level, cannot be directly applied to SimulST. Integration of a segmentation

---

*Equal contribution. Work was done during Linlin Zhang's internship at DAMO Academy, Alibaba Group.

†Corresponding author.

module into SimulST has become a common practice, as it serves as a prerequisite for resembling SimulMT policies. Notable examples include the connectionist temporal classification (CTC)-based approach (Ren et al., 2020; Yang et al., 2023) and the fixed-sized chunk method (Ma et al., 2021). However, CTC was not designed for simultaneous scenarios and the fixed-sized chunk method lacks flexibility in representing semantic meanings. The Continuous Integrate-and-Fire (CIF) method (Dong and Xu, 2020), which is ideally suited for streaming input, has recently attracted more attention in the streaming speech-to-text area (Dong et al., 2022a; Shu et al., 2023; Fu et al., 2023). Our work builds upon the research line of the CIF module.

We propose a fine-grained CIF module as the traditional CIF lacks clear and interpretable training supervision, as shown in Figure 1. We use the Montreal Forced Aligner (MFA) (McAuliffe et al., 2017) to acquire frame-word or -phoneme alignments, and use them to pre-train an acoustic encoder with a fine-grained CIF loss. By aligning speech and transcription, we can readily adapt the read and write policy of SimulMT to SimulST.

An issue similar to the exposure bias (Ranzato et al., 2016), whereby the prediction error of segmentation may cause a gap between training and inference, is rarely discussed in previous works. We propose a robust-alignment training strategy to minimize this gap. Moreover, previous SimulST works based on Wav2Vec2.0 (Baevski et al., 2020) often overlook its bidirectional attention, which isn't inherently a streaming fashion. We discuss this by suggesting a random wait-$k$ policy, making our model compatible with a bidirectional encoder. Our robust strategy solely involves frame-transcription alignment and is independent of the read/write policy, thus making it feasible to combine with other adaptive policies. Our main contributions can be summarized as follows.

(**1**) We designed a cross-modal pre-training method for the acoustic encoder, based on our fine-grained CIF module. This approach allows for the segmentation and alignment of speech with text representation in both offline and online scenarios.

(**2**) We enhanced the robustness and introduced more randomness to the wait-$k$ policy for real-time speech translation. Our robust-alignment training strategy can effectively minimize the gap between training and inference.

(**3**) We carried out experiments across multiple language directions on the MuST-C dataset. Our simultaneous translation model achieved a better trade-off between translation quality and various latency requirements with a single model, in both restricted and unrestricted scenarios.

## 2 Related Works

Speech translation can be roughly classified into offline and simultaneous scenarios, with our discussion primarily focusing on end-to-end models.

**Offline Speech Translation** generates each target token based on the full audio representation. Following the advent of the first end-to-end neural network model for ST (Berard et al., 2016), a significant portion of end-to-end ST works emphasized offline scenarios. Weiss et al. (2017); Berard et al. (2018); Bansal et al. (2019); Alinejad and Sarkar (2020); Dong et al. (2021) demonstrated the effectiveness of pre-training in improving performance. Consequently, this has become the standard paradigm for current end-to-end models. Other prevalent research areas include multitask learning (Bahar et al., 2019; Liu et al., 2020; Indurthi et al., 2020; Han et al., 2021; Ye et al., 2021; Zhang et al., 2023), knowledge distillation (Liu et al., 2019; Gaido et al., 2020; Wang et al., 2023), and curriculum learning (Kano et al., 2018; Wang et al., 2020).

**Simultaneous Speech Translation** was first realized through a pipelined system featuring streaming ASR and a SimulMT model. ASR played a natural role in segmenting speech frames and aligning with transcriptions, albeit at the cost of increased latency. Recently, end-to-end SimulST has attracted more research attention. Thus, a SimulST model requires a policy that determines whether to wait for more speech frames or generate new target tokens. Previous works often used a fixed size speech chunk (Ma et al., 2020b; Nguyen et al., 2021; Ma et al., 2021; Liu et al., 2021).

Adaptive size policy has been thoroughly explored in SimulMT (Arivazhagan et al., 2019; Ma et al., 2020a; Zhang et al., 2020). As to SimulST, the works of (Dong et al., 2022a; Zhang et al., 2022) are most relevant to our research. The former first applied CIF to SimulST, and the latter determined the read/write policy by learning to segment audio frames into meaningful units. Our model achieves cross-modal token-level alignment through pre-training a novel fine-grained CIF mod-

ule, thereby facilitating downstream SimulST. Considering the frame length of each segmentation, our model also triggers an adaptive policy.

## 3 Main Method

End-to-end simultaneous speech translation models typically employ a sequence-to-sequence learning framework built on an encoder-decoder architecture. For a standard ST training corpus, $\mathcal{D} = \{(\mathbf{s}, \mathbf{x}, \mathbf{y})\}$ represents a triplet dataset that includes audio, transcription, and translation sequences. The SimulST model is generally defined as a probabilistic model:

$$p(\mathbf{y}|\mathbf{s}; \theta) = \prod_j p(y_j | \mathbf{s}_{\leq g(j)}, \mathbf{y}_{<j}; \theta), \quad (1)$$

where $\theta$ is the model parameters, and $g(j)$ is a monotonically non-decreasing function that indicates the ending timestamp of the audio required to generate the $j$-th target token.

In our work, the aim of the pre-training stage is to embed audio features into the text representation space (*i.e.*, source embedding space) and to identify the candidate ending timestamp for the function $g(j)$. The fine-tuning stage directly optimizes the SimulST model as per Eq. (1). To achieve these objectives, we initially utilize the transcription-translation pairs in $\mathcal{D}$ to pre-train an offline machine translation model (NMT). In practice, additional MT corpus is more readily available, allowing us to pre-train the NMT using even larger data sets. The overall model architecture, comprising both stages, is displayed in Figure 2.

### 3.1 Semantic Alignment Pre-Training

In this stage, the primary objective is to achieve cross-modal semantic alignment between speech and text. We utilize only the audio-transcription pairs $(\mathbf{s}, \mathbf{x})$ to train the parameters of the acoustic encoder, while temporarily discarding the NMT encoder and decoder.

### 3.1.1 Fine-grained CIF Supervision

Similar to recent SimulST works with adaptive size policy (Dong et al., 2022a; Zhang et al., 2022), we adopt Wav2Vec2.0 (Baevski et al., 2020) as the raw speech feature extractor. For an input audio $\mathbf{s}$, we denote the output features of Wav2Vec2.0 as acoustic tokens $\mathbf{a} = (\boldsymbol{a}_1, \boldsymbol{a}_2, ..., \boldsymbol{a}_T)$.

We also utilize the Montreal Forced Aligner (MFA) to achieve frame-word alignment, as shown in Figure 1. Because Wav2Vec2.0 involves a 320x subsampling, the alignment between acoustic tokens $\mathbf{a}$ and the word-level transcription is also available. We can express the acoustic segment sequence as,

$$(\boldsymbol{a}_1, \boldsymbol{a}_2, ..., \boldsymbol{a}_T) = (\mathbf{a}_{b_1:e_1}, ..., \mathbf{a}_{b_W:e_W}). \quad (2)$$

where $W$ is the number of words in $\mathbf{x}$, and $b_\omega, e_\omega$ indicate the beginning and ending indices that align to the $\omega$-th word. Note that $\mathbf{x}$ is usually represented as a subword sequence $(x_1, ..., x_L)$ according to the NMT tokenizer, so $L \neq W$ in general. For our running example, $b_4, e_4$ are aligned to the word "motorcade", whereas "motorcade" in $\mathbf{x}$ is represented as subwords $x_4$ "motor" and $x_5$ "cade". Nevertheless, we can easily determine how many subwords each segment corresponds to and denote it as $(l_1, ..., l_W)$, satisfying $L = \sum_{\omega=1}^{W} l_\omega$. This quantity will serve as the fine-grained supervision for our proposed CIF.

Following the traditional CIF, we obtain the CIF weights $\hat{\boldsymbol{\alpha}} = (\hat{\alpha}_1, ..., \hat{\alpha}_T)$ by inputting the acoustic tokens into a linear layer with sigmoid activation. Contrasting with the conventional loss in CIF, we utilize labels from forced alignment and propose a novel fine-grained CIF loss, defined as follows.

$$\mathcal{L}_{\mathrm{cif}} = \sum_{\omega=1}^{W} \left| \sum_{t=b_\omega}^{e_\omega} \hat{\alpha}_t - l_\omega \right| + \left| \sum_{t=1}^{T} \hat{\alpha}_t - L \right| \quad (3)$$

The first term is our proposed fine-grained subword-level boundary loss, where the learning signal corresponds to the number of subwords within a given timestamp interval. The second term is the original CIF loss that learns the subword boundary under a weak signal – the total number of subwords.

### 3.1.2 Semantic Alignment Supervision

During early training that Eq. (3) does not converge, the $\hat{\boldsymbol{\alpha}}$ is ill-defined for boundary inference, typically necessitating length normalization.

$$\boldsymbol{\alpha} = L \cdot \frac{\hat{\boldsymbol{\alpha}}}{\sum_t \hat{\alpha}_t} \quad (4)$$

With this update, $\boldsymbol{\alpha}$ fulfills the condition $\sum_t \alpha_t = L$. During inference, normalization is not required, as the value of $L$ is unknown. However, it is theoretically reasonable, as the loss $\mathcal{L}_{\mathrm{cif}}$ in Eq. (3) converges to 0, $\sum_t \hat{\alpha}_t$ will also converge to the underlying $L$.

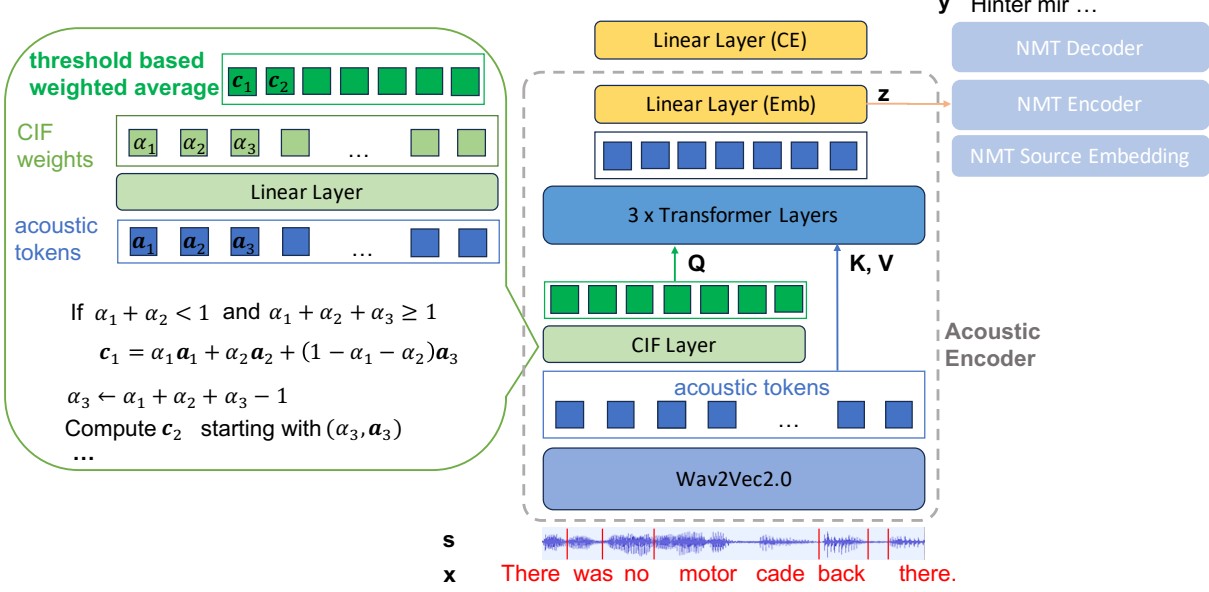

Figure 2: The overall model architecture of the proposed approach. The NMT is firstly pre-trained. In the pre-training stage for the acoustic encoder, the NMT encoder and decoder is temporarily discarded. In the fine-tuning stage for SimulMT task, the model is trained end-to-end. In the left panel, an example illustrates the threshold based weighted average.

Once we have the normalized weights, we process the acoustic tokens using a threshold-based weighted averaging from left to right, suitable for the simultaneous scenario. An intuitive example featuring $\alpha_{1,2,3}$ is depicted in the left panel of Figure. 2 (for more details, refer to Appendix A.1). During training, the averaged acoustic tokens should possess the same subword length as the transcription $\mathbf{x}$, denoted as $\mathbf{c} = (\boldsymbol{c}_1, ..., \boldsymbol{c}_L)$.

Next, we incorporate several Transformer layers for semantic alignment learning. In contrast to the traditional multi-head attention (MHA) of the encoder, our approach is designed as follows.

$$\mathbf{h}_i = \text{MHA}(Q = \mathbf{h}_{i-1}, K = \mathbf{a}, V = \mathbf{a}). \quad (5)$$

where $\mathbf{h}_0$ is initialized as $\mathbf{c}$. The output length for each layer will always be the same as that of $\mathbf{c}$.

For supervisory learning, we propose two semantic objectives through two linear layers, as shown in Figure 2. The first linear layer maps $\mathbf{h}$ to the source embedding space of NMT via $\mathbf{z} = \text{Linear}_{\text{emb}}(\mathbf{h})$. The corresponding loss aims to align the acoustic features with subword embeddings. The second linear layer further maps the output $\mathbf{z}$ of the acoustic encoder to the discrete vocabulary, employing

Cross-Entropy (CE) loss.

$$\mathcal{L}_{\text{emb}} = \|\mathbf{z} - \text{Emb}(\mathbf{x})\|_2, \quad (6)$$

$$\mathcal{L}_{\text{ce}} = \sum_{l=1}^{L} \text{CE}\left(\text{Linear}_{\text{ce}}(\boldsymbol{z}_l), x_l\right). \quad (7)$$

It is worth noting that the mixup (Chen et al., 2021; Fang et al., 2022b) between $\mathbf{z}$ and $\text{Emb}(\mathbf{x})$ is randomly applied when calculating the CE loss. Thus, our final pre-training loss is the aggregation of Eq. (3,6,7), *i.e.*,

$$\mathcal{L}_{\text{pt}} = \mathcal{L}_{\text{cif}} + \lambda \cdot \mathcal{L}_{\text{emb}} + \mathcal{L}_{\text{ce}}. \quad (8)$$

where a tunable hyper-parameter $\lambda$ is multiplied to balance the scale of mean squared loss $\mathcal{L}_{\text{emb}}$. Note that the embedding matrix is frozen, and the linear layer for CE will be discarded for ST inference.

### 3.2 Robust and Random Wait-$k$-tokens

After the previous stage, both the acoustic encoder and the NMT model are well pre-trained. The output of the acoustic encoder is expected to align with the source embedding space in terms of both semantic meaning and sequence length. During the fine-tuning stage, we will use the full ST data, denoted as $\mathcal{D}$, to train the end-to-end SimulST model.

#### 3.2.1 Random Wait-$k$

Our proposal draws inspiration from the efficient wait-$k$ strategy (Elbayad et al., 2020), originally

designed for SimulMT with a unidirectional encoder. However, the raw speech feature extractor, Wav2Vec2.0, is a bidirectional encoder. Prior SimulMT studies (Dong et al., 2022a; Zhang et al., 2022) directly utilized it and overlooked the discrepancy between offline and streaming input. We propose a robust and random training strategy to address this issue.

In particular, we first sample a source word-level length $\omega$ from a uniform distribution $\mathcal{U}(1, W)$, and sample a $k \in \mathcal{U}(K_1, K_2)$ as the number of waiting timestamps, where $K_1 < K_2$. According to the MFA alignment and word-level length $\omega$, we can readily derive the subword-level source length $l_s \in [1, L]$. The target length then becomes $l_t = l_s - k + 1$. In our case, $K_1$ could be a negative number as long as $l_t$ does not exceed the translation sentence length. Then, we propose the random wait-$k$-tokens loss.

$$\mathcal{L}_{\text{wk}}^{\text{st}} = \sum\nolimits_{l_s \in \mathcal{S}} \log p(y_{l_t} | \mathbf{z}_{\leq l_s}, \mathbf{y}_{<l_t}), \quad (9)$$

where $\mathcal{S}$ represents a sampled set of source lengths. To improve efficiency, we use the same $k$ for each example, but resample $k$ across examples in a batch. Note that **(i)** our approach permits bidirectional attention on the partial input, as $\mathcal{L}_{\text{wk}}^{\text{st}}$ is essentially optimizing the sampled prefix pairs in the wait-$k$ decoding path; **(ii)** However, we also encounter the unresolved issue from (Dong et al., 2022a; Zhang et al., 2022): during inference, the encoder hidden states of Wav2Vec2.0 cannot be cached for streaming input. We will discuss potential solutions to this problem as future work in the limitations section.

### 3.2.2 Robust Wait-$k$

As previously mentioned in Eq. (4), unless the loss $\mathcal{L}_{\text{cif}}$ is strictly equal to 0, the inferred $\hat{\alpha}$ will be suboptimal, and a discrepancy will exist between $\hat{L} = \sum \hat{\alpha}$ and the actual transcription length. To address this issue, we propose a robust training strategy. Essentially, by emulating the scheduled sampling (Bengio et al., 2015) that can mitigate exposure bias, we use the predicted alignment for the loss Eq. (9) with a probability (*e.g.*, 0.5).

Specifically, when sampling the source length in $\mathcal{L}_{\text{wk}}^{\text{st}}$, we randomly select the sampling rule from the following two methods.

1. Sample $\omega \sim \mathcal{U}(1, W)$, then derive $l_s$;

2. Directly sample $l_s \sim \mathcal{U}(1, \hat{L})$.

If the former distribution is selected, it implies that the training can utilize the ground-truth segmentation based on the MFA, otherwise, the sampled length from the latter distribution is already at the subword-level and the training uses the predicted segmentation based on $\hat{\alpha}$.

### 3.2.3 Overall Loss

The overall loss of the fine-tuning stage is defined to optimize all model parameters.

$$\mathcal{L}_{\text{ft}} = \mathcal{L}_{\text{off}}^{\text{st}} + \mathcal{L}_{\text{wk}}^{\text{st}} + \mathcal{L}_{\text{off}}^{\text{mt}} + \mathcal{L}_{\text{wk}}^{\text{mt}} + \mathcal{L}_{\text{pt}} \quad (10)$$

where $\mathcal{L}_{\text{off}}$ is the offline translation loss conditional on full speech or transcription, and $\mathcal{L}_{\text{wk}}^{\text{mt}}$ is defined via $p(y_{l_t} | \mathbf{x}_{\leq l_s}, \mathbf{y}_{<l_t})$.

## 4 Experiments

### 4.1 Experimental Setting

#### 4.1.1 Dataset

For a fair comparison with previous works, we conduct our experiments on the widely used **MuST-C V1**: English→{German, French, Spanish} (En→{De, Fr, Es}) (Gangi et al., 2019). For our multitask ST model with auxiliary MT data, we extract 20M En-Fr and En-Es 15M sentence pairs from the WMT14 and WMT13 data. For En-De, we use all of the WMT14 dataset. The data statistics are shown in Table 1.

| corpus | ST(Hrs/Sents) | Optional MT(Sents) |
|--------|---------------|--------------------|
| En-De  | 408/234K      | 4.5M(WMT14)        |
| En-Fr  | 492/280K      | 20M(WMT14)         |
| En-Es  | 504/270K      | 15M(WMT13)         |

Table 1: The statistics for the three language pairs.

Following the preprocessing recipes of STEMM (Fang et al., 2022b), we use MFA (McAuliffe et al., 2017) to process ASR data to obtain speech-text alignment information. We also filter out the raw audio larger than 450000 frames. The text vocabulary consists of 10,000 subword units learned by SentencePiece (Kudo, 2018), shared between the source and target languages.

#### 4.1.2 Model Configuration and Evaluation

In all experiments, our semantic alignment module comprises 3 transformer layers. The Neural Machine Translation (NMT) model employs 6 transformer encoder and decoder layers. Both have a hidden dimension of 512 and utilize eight attention heads. We pre-train two variants of NMT models: one using the MuST-C data exclusively, and another incorporating an additional MT corpus. During SimulST training, we sample $k$ between 3 and

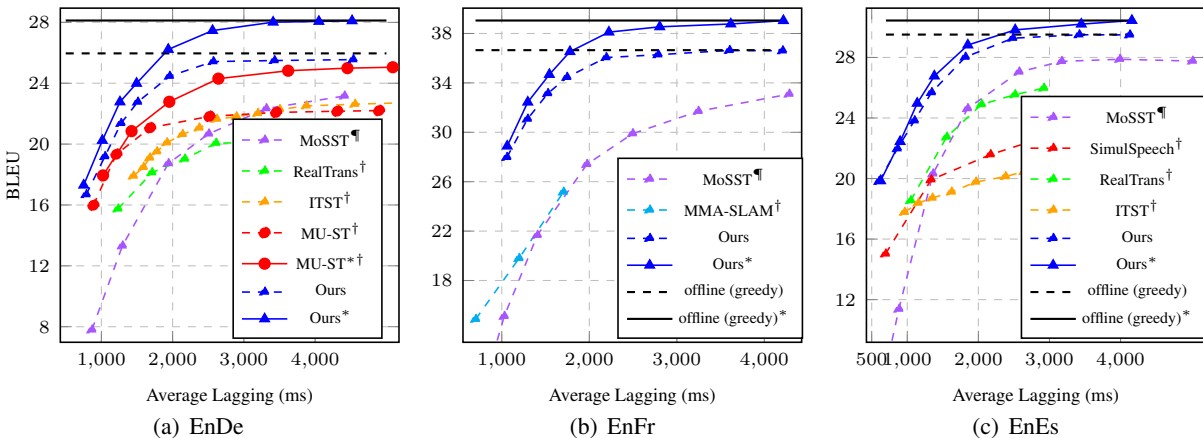

Figure 3: The translation quality (BLEU) against the latency metrics (AL) on the tst-COMMON set of MuST-C En-De, En-Fr and En-Es dataset. † denotes that the results are obtained from corresponding papers. ¶ denotes that the results are based on our re-implementation, where our implemented MoSST is slightly better than the original results and includes the results on large latency region. All dashed lines are models trained in constrained scenarios. * denotes that the results of solid lines are from models that utilize external machine translation data.

10, while for evaluation, $k$ ranges from 2 to 25. We conduct our training using 8 V100 GPUs, with a batch size per device of 3.2M audio frames.

Model selection is based on the corresponding development set, and we report final results on the tst-COMMON set. We utilize the detokenized case-sensitive BLEU score, calculated using sacre-BLEU[1]. Following the convention in the SimulST, we use the average lagging (AL) evaluation model to measure the latency (Ma et al., 2019b) based on the toolkit SemiEval[2].

### 4.2  Main Results of SimulST

We compared our model with recent simultaneous ST models: SimulSpeech (Ren et al., 2020), RealTranS (Zeng et al., 2021), MoSST (Dong et al., 2022b), MU-ST (Zhang et al., 2022), MMA-SLM (Indurthi et al., 2022), and ITST (Zhang and Feng, 2022). Most of these works explored the SimulST task using the MuST-C dataset alone. MU-ST also conducted experiments with external data, using both additional Automatic Speech Recognition (ASR) and Machine Translation (MT). In our case, we did not use additional ASR data.

**Quality and latency trade-off by single model** In Figure 3, we plot the BLEU *v.s.* AL curve of a single checkpoint of our ST model for both offline and online scenarios across different latency. To achieve these results, we leverage our proposed wait-$k$-tokens strategy during inference, with token

boundaries based on CIF prediction.

Regarding En-De, which is the most researched language pair, our method significantly outperforms all previous approaches in terms of translation quality at high latency. In the constrained scenario (represented by dashed lines), our method scores at least 2 BLEU higher than MU-ST and MoSST, even outperforming MU-ST in the unconstrained case. In the unconstrained scenario, our method can further enhance performance. At low latency, it seems the curves of our approach and MU-ST are close. However, if we focus on the snapshot around latency 1000ms, our method surpasses MU-ST by approximately 2 BLEU. For En-Fr, fewer works have explored this language pair. Compared to existing baselines, our model in a constrained scenario outperforms them by a significant margin. For instance, around low latency (1000ms), our method achieves a BLEU score exceeding 28. In the case of English to Spanish (En-Es), although our results are generally better, we found the additional MT corpus appears to be less beneficial. The exact numerical results in Figure 3 can be found in Appendix A.7.

**Analysis of Pre-training Stage** In the pre-training stage, the loss $\mathcal{L}_{pt}$ primarily optimizes semantic alignment learning in terms of both sequence length and semantic representations. We will illustrate the performance of these objectives.

First, we conducted a statistical analysis comparing the number of predicted segments by CIF to the actual subword lengths of the corresponding tran-

[1] https://github.com/mjpost/sacrebleu
[2] https://github.com/facebookresearch/SimulEval

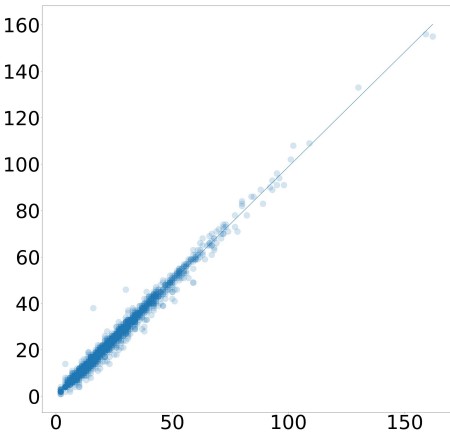

Figure 4: A linear regression analysis using the ground-truth length (y) and the predicted length (x) on En-De tst-COMMON dataset.

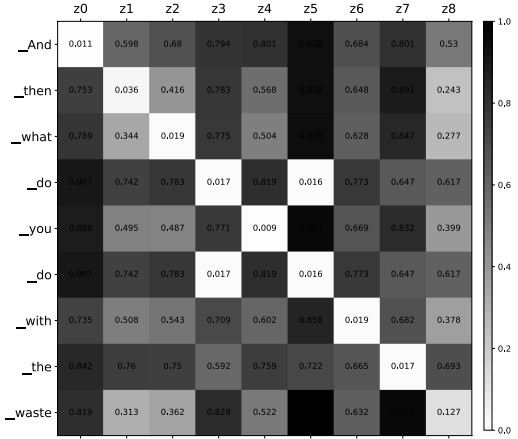

Figure 5: An example of the alignment for semantic representation via $L_2$ distance. $\mathbf{z}$ is the acoustic encoder output after $\text{Linear}_{\text{emb}}$ as shown in Figure 2.

scriptions. Figure 4 demonstrates the En-De test set with the ground-truth length (y) and the predicted length (x) by our alignment module used to fit a linear regression. The regression line closely aligns with the diagonal of the coordinate system, *i.e.*, $y = x$, indicating the high accuracy of our length prediction. Furthermore, few outliers are distant from the regression line, suggesting that our length prediction exhibits low bias and variance. These results collectively highlight the effectiveness of our speech pre-training in predicting sentence length. The analysis of length prediction for En-Fr and En-Es is provided in Appendix A.2.

Second, Figure 5 illustrates the $L_2$ distance between the acoustic encoder output and the word embeddings of its corresponding transcription. The elements along the diagonal generally exhibit smaller values, indicating a closer semantic distance for cor-

| Model | En-De | En-Fr | En-Es |
|---|---|---|---|
| RealTranS (Zeng et al., 2021) | 23.0 | - | - |
| MoSST (Dong et al., 2022b) | 24.9 | 35.3 | - |
| XSNET (Ye et al., 2021) | 25.5 | 36.0 | 29.6 |
| STEMM (Fang et al., 2022b) | 25.6 | 36.1 | 30.3 |
| ConST (Ye et al., 2022) | 25.7 | 36.8 | **30.4** |
| Ours | **26.8** | **37.7** | **30.4** |
| XSNET (Ye et al., 2021) | 27.8 | 38.0 | 30.8 |
| STEMM (Fang et al., 2022b) | 28.7 | 37.4 | 31.0 |
| ConST (Ye et al., 2022) | 28.3 | 38.3 | 32.0 |
| STPT (Tang et al., 2022) | **29.2** | **39.7** | **33.1** |
| Ours* | **29.2** | **39.7** | 31.0 |

Table 2: The performance of the offline ST on the tst-COMMON dataset. The beam size of 5 is adopted in our evaluation. The top of the table represents the results using the constrained ST data, while the bottom represents the results using external data. The amount of external data varies a lot between individual models.

responding acoustic and text representations. If the transcription has two same tokens (such as "do" in Figure 5), we can still observe semantic similarity for their corresponding acoustic output ($z_3$ and $z_5$ in Figure 5). This phenomenon further verifies the effectiveness of semantic learning. Additional visualizations can be found in Appendix A.3.

Third, we explicitly visualize the audio-text alignment in Figure 6. In the provided example, we can observe there is a strong correlation between the $\alpha$ values in the second row and the amplitude values of the speech in the first row. The segmentation activations in the third row are also highly correlated with the corresponding positions of the subwords. For meaningless blank characters without a corresponding meaningful subword, the predicted $\alpha$ is almost 0, and no subword is generated. Additional examples can be found in Appendix A.4.

### 4.3 Main Results of Offline ST

As the bidirectional architecture of Wav2Vec2.0, our ST model is naturally compatible with offline speech translation. Specifically, we compare our model, previously used for SimulST in Figure 3, with recently published offline ST methods: XS-NET, STEMM, ConST, and STPT. We also include two earlier SimulST works (RealTranS and MoSST) that evaluate both offline and online tasks.

The overall results are summarized in Table 2. In data-constrained scenarios, our offline ST model's BLEU evaluation matches the performance of these competitive offline models. In unconstrained scenarios, only the performance of En-Es exhibits a

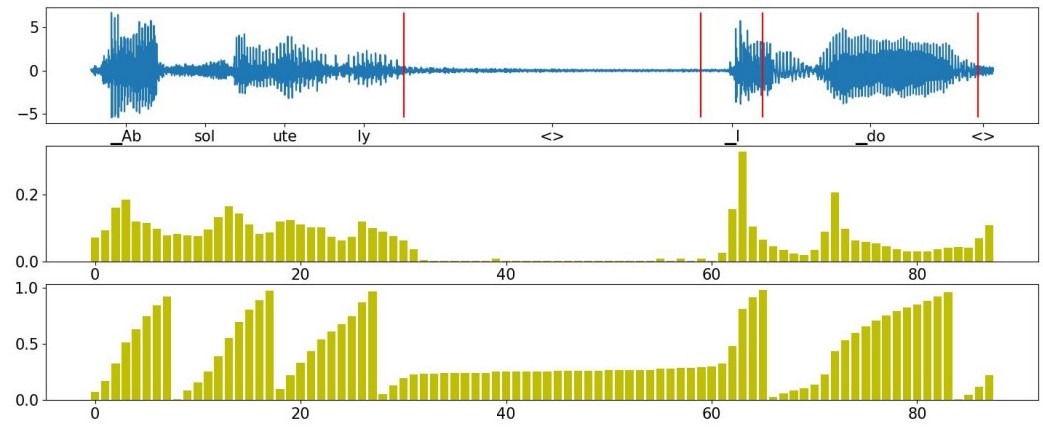

.

Figure 6: **First Row**: word-level segmentation of the audio by MFA and the corresponding subwords in each segmentation. **Second Row**: predicted $\alpha$ by fine-grained CIF module. **Third Row**: the accumulated summation of $\alpha$, which will be reset once exceeding threshold 1.0.

| model | $R^2$ wait-$k$-tokens | Fine-grained CIF loss | CE loss | pre-train | beam=5 |
|---|---|---|---|---|---|
| (a) | √ | √ | √ | √ | 29.15 |
| (b) | | √ | √ | √ | 29.00 |
| (c) | √ | | √ | √ | 28.70 |
| (d) | √ | √ | √ | | 27.95 |
| (e) | | | √ | √ | 28.62 |
| (f) | | | | √ | 28.74 |
| (g) | | | | | 27.63 |

Table 3: The BLEU scores of offline ST from the ablation experiments conducted on tst-COMMON En-De. "$R^2$ wait-$k$" means the robust and random wait-$k$ tokens training. "Fine-grained CIF" indicates the use of fine-grained CIF loss (first term in Eq. (3)) of both pretraining and fine-tuning. "CE loss" refers to the use of CE loss during fine-tuning. "pre-train" indicates the use of our proposed semantic alignment pre-training stage. The pre-trained NMT is always required.

significantly lower BLEU score. We hypothesize the main reason for this is the considerable variation in the volume of external MT data across different models. Our selected 15M En-Es MT corpus may also have a potential domain mismatch with MuST-C. Generally, these results demonstrate that our single checkpoint can maintain robust offline translation quality while also adapting to real-time translation requirements.

### 4.4 Ablation Studies

We conducted ablation experiments on the unconstrained En-De dataset. The different settings and critical offline evaluations of our control experiments are listed in Table 3. The corresponding BLEU *v.s.* AL curves are shown in Figure 7.

Regarding the offline task, the $R^2$ wait-$k$-tokens training did not yield any notable difference when comparing model (a) and (b), however, in the SimulST task, the absence of this strategy changed the curve from an arc shape to an almost straight line. When the fine-grained CIF loss was removed (model (c)), the offline performance decreased by about 0.5 BLEU, while the arc curve also uniformly shifted downwards by 0.5 BLEU. Removing the entire semantic alignment pre-training (model (d)) resulted in SimulST performance at low latency that was on par with model (c), but the BLEU scores at high latency significantly decreased. In summary, $R^2$ wait-$k$ primarily enables streaming; the semantic alignment pre-training improves the performance at high latency; and the fine-grained CIF loss is the final touch that further enhances performance at low latency. Another two loss related ablation studies can be found in Appendix A.5. The analysis of how we design the current pre-training loss can refer to Appendix A.6.

### 4.5 Stable Inference at Low Latency

In practice, during the inference stage, the first few target tokens often have low quality when the streaming input audio is short. Various strategies have been proposed by previous works to address this problem. *E.g.*, RealTranS employs the Wait-K-Stride-N strategy, while MU-ST uses a tail-truncation trick. In our approach, we empirically find that when the first few short speech segments are processed as streaming input, the CIF weights tend to bias towards larger values and require more

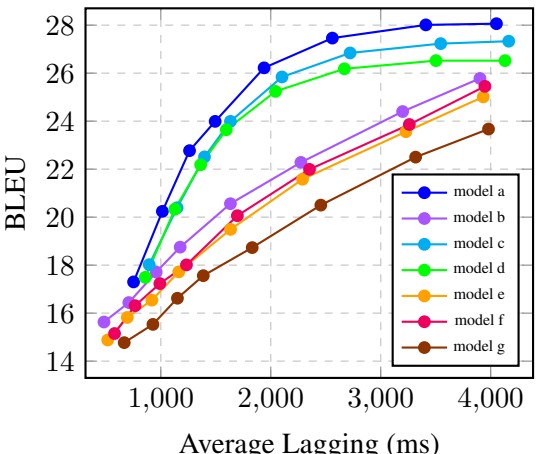

Figure 7: Ablation experiments conducted on simultaneous translation corresponding to Table 3.

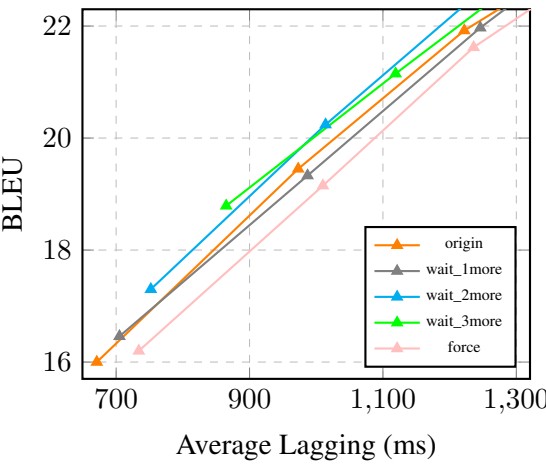

Figure 8: **origin** means no additional trick is applied. **wait_$n$more** is to wait for more $n$ extra segments before decoding the first target token, while the other decoding is normal. **force** is to force the model not to finish the hypothesis before the source audio is fully read.

input frames to stabilize. Conversely, with longer speech inputs, the decoding process may end prematurely. Therefore, we propose a strategy to wait for a few extra segments before initiating the first write operation. After this, the read and write behavior returns to normal. According to our analysis in Figure 8, our "wait-$n$more" strategy effectively improves performance at low latency, and we apply the wait-2more strategy in our experiments.

## 5  Conclusion

We have proposed a two-step training method for simultaneous speech translation. The first step involves performing a token-level cross-modal alignment between the audio and the transcription in terms of both sequence length and semantic rep-

resentation. The second step entails end-to-end training of the ST model, leveraging a robust and random wait-$k$ tokens policy. Remarkably, our model manages to satisfy the objectives of offline and streaming speech translation within a single checkpoint. Experimental results substantiate that our approach achieves a commendable trade-off between translation quality and latency. Given that our semantic alignment pre-training is independent of the downstream policy in simultaneous translation, we aim to explore opportunities to better accommodate an adaptive read/write policy in future work.

## Limitations

As discussed in Section 3, our training strategy can indeed mitigate some issues of the bidirectional Wav2Vec2.0, but it still encounters the same unresolved issue as in (Dong et al., 2022a; Zhang et al., 2022). During inference, the encoder hidden states of Wav2Vec2.0 cannot be cached for streaming input. Thus, it is not an efficient audio feature extractor in a streaming scenario since the recalculation of audio features is necessary whenever new streaming input arrives. A potential research direction could be replacing Wav2Vec2.0 with a streaming acoustic encoder. It's worth noting that Wav2Vec2.0's self-supervised learning is pre-trained by masking 49% of all time steps with a mean span length of 300ms. Therefore, finetuning the existing Wav2Vec2.0 with streaming input should be a feasible approach.

## Ethics Statement

Our work complies with the ACL Ethics Policy. Our experiments are based on the open-sourced dataset that is widely used in academia, and there is no violation of this dataset. Our writing is completely based on the authors without plagiarism.

## Acknowledgements

We would like to thank all the anonymous reviewers for their insightful and helpful comments. This work was supported by Alibaba Group through Alibaba Research Intern Program.

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

## A  Appendix

### A.1  Threshold-Based Weighted Averaging of CIF

In the section, we formulate the threshold-based weighted averaging of CIF into Algorithm 1.

### A.2  Visualization of Length Prediction for CIF

We conducted linear regression analysis on the predicted length of CIF and actual transcript token

---

**Algorithm 1** Threshold-based Weighted Averaging in Continuous Integrate-and-Fire (CIF)

**Input:** The output features of Wav2Vec2.0 as acoustic tokens $\mathbf{a} = (\boldsymbol{a}_1, \boldsymbol{a}_2, ..., \boldsymbol{a}_T)$. The normalized CIF weights $\boldsymbol{\alpha} = (\alpha_1, \alpha_2, ..., \alpha_T)$ satisfying $\sum_{t=1}^{T} \alpha_t = L$. Threshold $\delta = 1$ be default.

**Output:** The averaged acoustic tokens after CIF module have the same subword length as the transcription $\mathbf{x}$, denoted as $\mathbf{c} = (\boldsymbol{c}_1, ..., \boldsymbol{c}_L)$.

1: **function** MAIN
2:     *// Initialize $i = 1$ and $i$ will end with $L$, initial accumulated weight $\alpha_0^a = 0$, initial accumulated state $\boldsymbol{a}_0^a = \mathbf{0}$;*
3:     $i = 1, \alpha_0^a = 0, \boldsymbol{a}_0^a = \mathbf{0}$
4:     **for** $t = 1 : T$ **do**
5:         *// calculate accumulated weight;*
6:         $\alpha_t^a = \alpha_{t-1}^a + \alpha_t$
7:         **if** $\alpha_t^a < \delta$ **then**
8:             *// no boundary is located;*
9:             $\boldsymbol{a}_t^a = \boldsymbol{a}_{t-1}^a + \alpha_t \times \boldsymbol{a}_t$
10:         **else**
11:             *// a boundary is located;*
12:             *// $\alpha_t$ is divided into two part, the first part $\alpha_{t_1}$ is used to fulfill the integration of current $c_i$;*
13:             $\alpha_{t_1} = 1 - \alpha_{t-1}^a$
14:             $\boldsymbol{c}_i = \boldsymbol{a}_t^a + \alpha_{t_1} \times \boldsymbol{a}_t$
15:             $i + +$
16:             *// The other part $\alpha_{t_2}$ is used for the next integration;*
17:             $\alpha_t^a = \alpha_{t_2} = \alpha_t - \alpha_{t1}$
18:             $\boldsymbol{a}_t^a = \alpha_{t_2} \times \boldsymbol{a}_t$
19:         **end if**
20:     **end for**
21:
22: **end function**
23: **return** $\mathbf{c} = (\boldsymbol{c}_1, ..., \boldsymbol{c}_L)$

---

length for different language pairs. From Figure 9, it can be seen that our method exhibits high prediction accuracy during both the cross-modal pretraining and ST fine-tuning phases. All models can fit the linear regression equation $y = x$. The data is primarily concentrated on this line. From the results, our model can maintain a high level of accuracy in predicting length whether or not fine-tuning is done on the ST. At the same time, there is little variation across different language pairs.

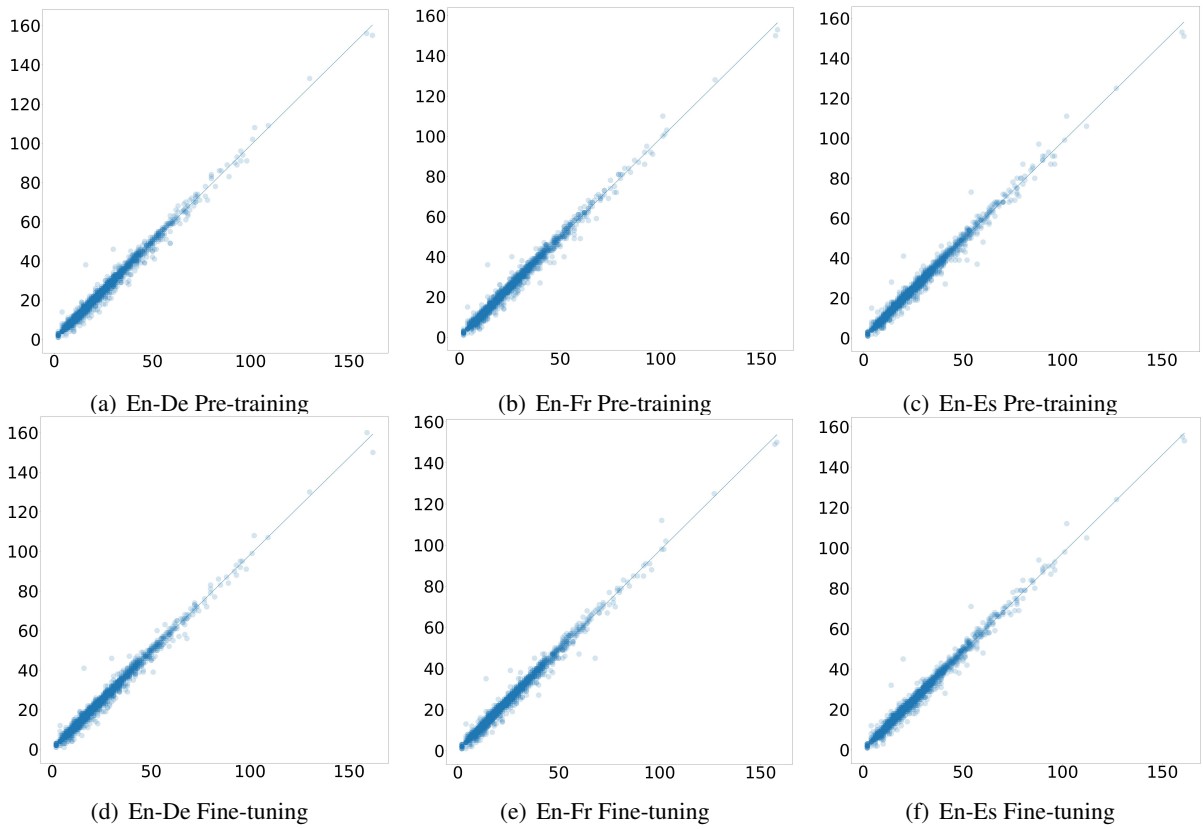

(a) En-De Pre-training      (b) En-Fr Pre-training      (c) En-Es Pre-training

(d) En-De Fine-tuning      (e) En-Fr Fine-tuning      (f) En-Es Fine-tuning

Figure 9: A linear regression analysis using the ground-truth length (y) and the predicted length (x) on En-De, En-Fr and En-Es tst-COMMON dataset. The first row corresponds to the cross-modal pre-training phase, while the second row represents the ST fine-tuning phase.

## A.3 Visualization of Cross-modal Alignment

We illustrate the $L_2$ distance between the acoustic encoder output and the word embeddings of its corresponding transcription in Figure 10. From Figure 10, it can be observed that regardless of the accuracy of length prediction, all examples exhibit lighter-colored areas along the diagonal, indicating that our method aligns speech and text distances more closely in that region, while distances are greater in other areas.

Additionally, subwords such as "y" in Figure 10(b), "to" in Figure 10(c), "_the" in Figure 10(d), and "_do" in Figure 10(e) are repeated, with their corresponding speech output being very similar. This suggests that our method not only simply brings speech and text closer in input position order, but truly surpasses input order to align them in semantic space.

## A.4 Speech Segmentation and Alignment

Our method, after the cross-modal align pre-training, is effective in achieving segmentation and alignment at the token-level. Figure 11 shows some

randomly selected examples, including complete speech sentences and some truncated sections of the speech. In general, we can observe an early activation for subwords, i.e., the accumulation sum of $\alpha$ tends to reach threshold 1.0 earlier than the boundary detected by MFA. We hypothesize that the boundary from MFA is usually located in the middle of consecutive silence frames, and the corresponding $\alpha$ for silence frame is almost 0, thus, the larger $\alpha$ values are usually predicted earlier.

## A.5 Additional Ablation on Loss

The three pre-training losses (CE loss, embed loss, and fine-grained CIF loss) are by default optimized in the fine-tuning stage, and the relevant ablation study is mainly conducted on the fine-tuning stage in Table 3. In this section, we conduct several new ablation studies of the auxiliary loss to supplement Table 3. In Table 4, if the loss is not marked as $\sqrt{}$, it means the corresponding loss is removed in both the pre-training and fine-tuning stages. Removing both the CE loss and embed loss in both the pre-training and fine-tuning stages results in

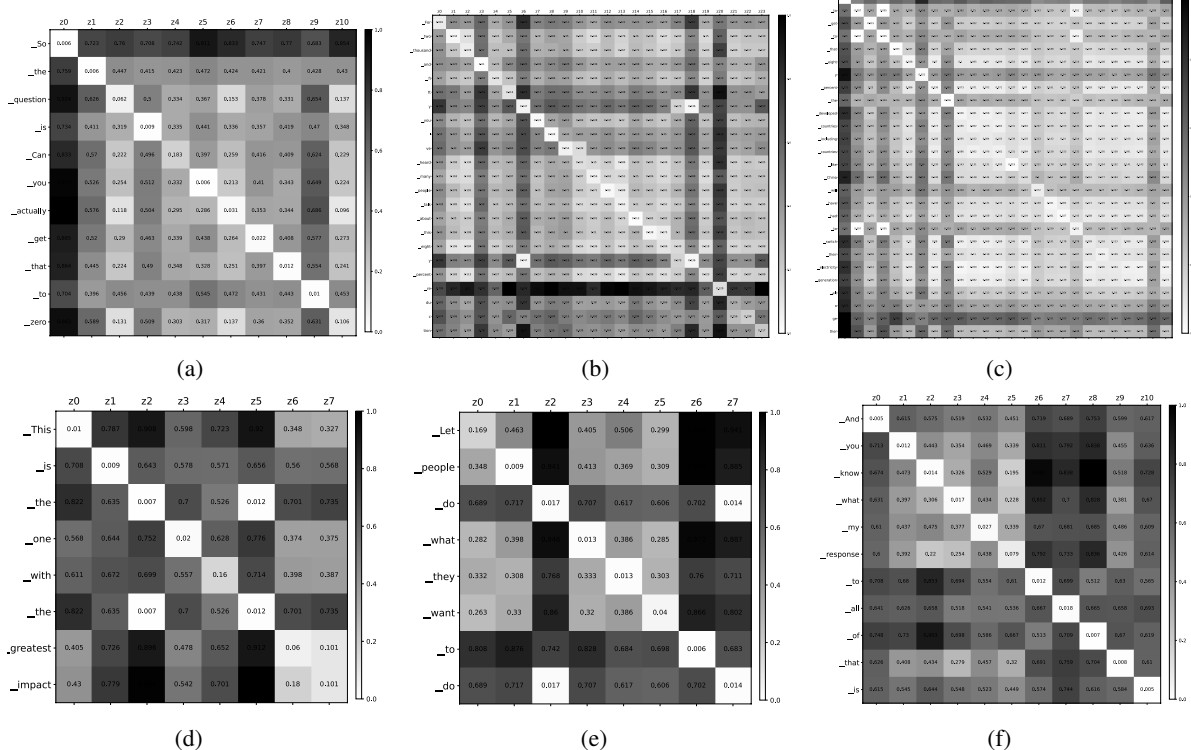

Figure 10: Some examples of speech segmentation and alignment on dev En-De, where **z** is the output of the acoustic encoder after Linear$_{emb}$ as shown in Figure 2. (Better to view by zoom-in) When a token appears more than once in a sentence, we can observe the lower valued elements may not appear on the diagonal, *e.g.*, "y" in (b), "to" in (c), "the" in (d), and "do" in (e).

a significant drop in performance. Based on the previous ablation experiment results, we can conclude that the CE loss plays a crucial role in the pre-training stage but has minimal impact in the fine-tuning stage. In Table 5, we conduct a new ablation study to explore whether pre-training loss $\mathcal{L}_{pt}$ is still required in the fine-tuning stage. In the bottom row of Table 5, after the pre-training stage, all three losses in $\mathcal{L}_{pt}$ are removed in fine-tuning. As expected, the BLEU on the dev set significantly dropped. Theoretically, the fine-grained CIF loss in is crucial because it provides the ability of audio segmentation for the streaming input. Without CIF loss, we found the length prediction becomes extremely worse, and imposes a big gap between the training and inference. In addition, the convergence becomes disordered. The convergence is usually achieved after 10+ epochs. Surprisingly, when $\mathcal{L}_{pt}$ is removed, the highest BLEU on the dev set is observed in the first epoch, followed by a decline in subsequent epochs. Therefore, our results indicate that $\mathcal{L}_{pt}$ is indeed crucial during the fine-tuning stage. Actually, including the $\mathcal{L}_{pt}$ in fine-tuning stage becomes common in speech translation, e.g.,

| model | Fine-grained CIF loss | CE loss | Emb loss | beam=5 |
|-------|------------------------|---------|----------|--------|
| (i)   | √                      | √       | √        | 29.15  |
| (ii)  | √                      | √       |          | 28.77  |
| (iii) | √                      |         |          | 27.84  |

Table 4: The BLEU scores of offline ST from the ablation experiments conducted on tst-COMMON En-De. If not marked with √, the corresponding loss is removed for **BOTH** pre-training and fine-tuning stages.

| pre-train loss | beam=5 | beam=1 |
|----------------|--------|--------|
| √              | 29.15  | 28.12  |
|                | 24.49  | 23.32  |

Table 5: Ablation study on pre-training loss in fine-tuning stage.

offline model STPT.

## A.6 The Ablation of Alignment Pre-training

We demonstrate ablation experiments for cross-modal alignment pre-training. As shown in Table 6, given sufficient training, there is little difference in accuracy and model length between the speech recognition models. However, in order to bring

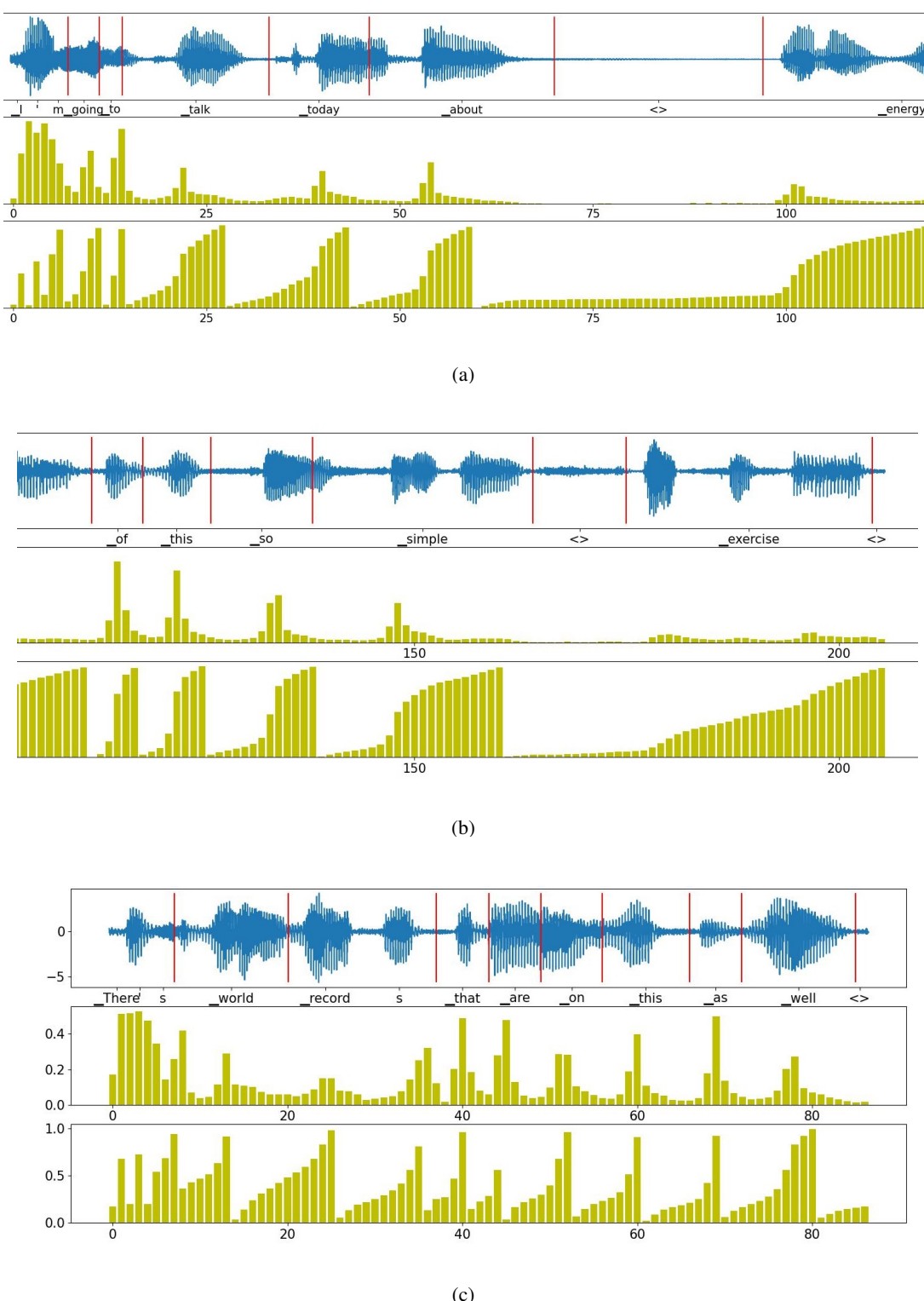

Figure 11: Some examples of speech segmentation and alignment on dev En-De. Each example is divided into three rows. The first row displays the word-level segmentation of speech computed by MFA and the tokens/subwords corresponding to each segment. The second row shows the alphas from CIF. The third row displays the accumulated scores, which activates the generation of tokens (when they exceed the threshold of 1.0).

| ce loss | ctc loss | embed | mixup | Accuracy | WER | Avg length bias ($L_2$) | Representation dist |
|:---:|:---:|:---:|:---:|:---:|:---:|:---:|:---:|
| √ | | √ | √ | 56.83 | 17.41 | 2.099 | 0.972 |
| | √ | | √ | 56.05 | 17.83 | 2.103 | |
| √ | | | √ | 56.28 | 18.01 | 2.102 | |
| √ | | √ | | 56.97 | 17.66 | 2.110 | 0.1275 |

Table 6: Ablation study on the alignment performance in pre-training stage. The scores obtained from the ablation experiments conducted on dev En-De. "ce loss" refers to the use of CE loss. "ctc loss" indicates the use of. "embed" indicates the use of MSE loss with the loaded MT embedding. "mixup" means the use of mixup between speech and text representations. "Avg length bias" shows the average $L2$ distance of the predict length and the groundtruth length. "Representation dist" meatures the distance between speech and text representations.

the word vectors of pre-trained MT closer, we ultimately chose the current combination.

From the first 3 metrics (accuracy, WER, and length prediction error), we cannot distinguish the importance of mixup. However, we define a new metric to measure the distance between speech and text representations called Representation dist,

$$\frac{1}{L_z} \sum_{i=1}^{L_z} \min_{j \in [1, L_e]} |\mathbf{z}_i - \mathbf{e}_j| \qquad (11)$$

Since the length of two representations may not be the same during inference, we select the minimum pairwise distance. In this metric, we can observe the mixup training can significantly reduce the representation distance.

### A.7 Numeric Results for Figures

These are the numerical results corresponding to our method's simultaneous ST, shown in Table 7

| Wait-k | 2 | 3 | 4 | 5 | 7 | 10 | 15 | 25 | Offline |
|--------|-----|-----|-----|-----|-----|-----|-----|-----|---------|
| Our | | | | | | | | | |
| BLEU | 16.68 | 19.18 | 21.36 | 22.76 | 24.46 | 25.42 | 25.49 | 25.55 | 25.96 |
| AL(ms) | 787 | 1049 | 1275 | 1509 | 1956 | 2569 | 3427 | 4532 | inf |
| Ours* | | | | | | | | | |
| BLEU | 17.30 | 20.24 | 22.77 | 23.99 | 26.22 | 27.46 | 28.01 | 28.10 | 28.12 |
| AL(ms) | 752 | 1014 | 1261 | 1494 | 1939 | 2560 | 3410 | 4518 | inf |

(a) En-De

| Wait-k | 2 | 3 | 4 | 5 | 7 | 10 | 15 | 20 | Offline |
|--------|-----|-----|-----|-----|-----|-----|-----|-----|---------|
| Our | | | | | | | | | |
| BLEU | 27.97 | 31.07 | 33.14 | 34.45 | 36.05 | 36.28 | 36.63 | 36.60 | 36.65 |
| AL(ms) | 1058 | 1294 | 1525 | 1744 | 2192 | 2786 | 3601 | 4208 | inf |
| Ours* | | | | | | | | | |
| BLEU | 28.88 | 32.44 | 34.68 | 36.53 | 38.11 | 38.54 | 38.77 | 39.04 | 39.05 |
| AL(ms) | 1062 | 1297 | 1544 | 1779 | 2224 | 2805 | 3617 | 4218 | inf |

(b) En-Fr

| Wait-k | 2 | 3 | 4 | 5 | 7 | 10 | 15 | 20 | Offline |
|--------|-----|-----|-----|-----|-----|-----|-----|-----|---------|
| Our | | | | | | | | | |
| BLEU | 19.81 | 21.97 | 23.82 | 25.68 | 28.02 | 29.27 | 29.50 | 29.49 | 29.51 |
| AL(ms) | 602 | 861 | 1100 | 1341 | 1816 | 2484 | 3417 | 4128 | inf |
| Ours* | | | | | | | | | |
| BLEU | 19.85 | 22.44 | 24.97 | 26.77 | 28.82 | 29.82 | 30.20 | 30.44 | 30.44 |
| AL(ms) | 625 | 900 | 1140 | 1381 | 1852 | 2518 | 3448 | 4159 | inf |

(b) En-Es

Table 7: Numeric results on MuST-C En-De, En-Fr, and En-Es tst-COMMON set. (Figure 3)