# OpenReview forum: "Training Simultaneous Speech Translation with Robust and Random Wait-k-Tokens Strategy"
_EMNLP/2023/Conference — EMNLP 2023 Main_

### Official Review · Reviewer_6LVe · 2023-08-02

**Soundness:** 4

**Excitement:**

4: Strong: This paper deepens the understanding of some phenomenon or lowers the barriers to an existing research direction.

**Missing References:**

I think that since SpeechUT is the current SOTA in offline ST, it should be included in Table 2 (second part / with external MT).

Ziqiang Zhang, Long Zhou, Junyi Ao, Shujie Liu, Lirong Dai, Jinyu Li, and Furu Wei. 2022. SpeechUT: Bridging Speech and Text with Hidden-Unit for Encoder-Decoder Based Speech-Text Pre-training. In Proceedings of the 2022 Conference on Empirical Methods in Natural Language Processing, pages 1663–1676, Abu Dhabi, United Arab Emirates. Association for Computational Linguistics.

**Paper Topic And Main Contributions:**

This paper addresses the challenges in Simultaneous Speech Translation (SimulST), a task designed to produce real-time translations of audio input. The authors identify the modality gap - a disparity due to differing word boundaries between audio and text - as a primary issue in the performance of offline speech translation and the application of simultaneous text translation (SimulMT) policies to SimulST.

To mitigate this issue, they propose the use of the Montreal Forced Aligner (MFA) and the utilization of audio transcription pairs for pre-training the acoustic encoder. This method facilitates a token-level cross-modal alignment, enabling the wait-k policy from SimulMT to adapt more effectively to SimulST.

Furthermore, the paper introduces a robust and random wait-k-tokens strategy to optimize the SimulST task. This approach enables a single model to address diverse latency requirements and reduces error accumulation of boundary alignment during inference, thereby enhancing the robustness and efficiency of the SimulST process.

Results in three standard language directions of MuST-C v1.0 indicate that the proposed method surpasses other SimulST methods, in both constrained and unconstrained scenarios (regarding external MT data), and across various latency conditions. Notably the same model is also very competitive in offline ST task. The authors carry out several ablation studies that shed light to the usefulness of each component in their model. Furthermore they showcase the ability of their method to accurately predict the length of text sequence from the acoustic information, as well as  the alignment between text and acoustic tokens.

**Questions For The Authors:**

A. What version of wav2vec 2.0 are you using? I assume it is the non-ASR fine-tuned, but please indicate it. Also if it is the base or large.

B. Did you carry out any ablation studies regarding the impact of mixup (Fang et al 2022) during the pre-training?

C. In L410-L411 it is stated that no additional ASR data are used. But doesn't the use of the external alignment tool indirectly implies the use of ASR data?

**Reasons To Accept:**

The paper is very well written and makes several contributions to the field of Simultaneous Speech Translation. The proposed pretraining strategy using audio-text alignment is effective in birding the modality gap, and making it easier to do transfer learning from SimulMT. It also seems to be able to accurately handle the sub-world level segmentation and the to predict the length of the text representation from the acoustic information of wav2vec2.0. Furthermore the proposed robust & random-k wait policy is integral to the success of the model as indicated by Figure 6, and manages to have high translation quality across different latencies. Overall, the paper would offer significant value to the NLP community both in terms of the novel techniques proposed and the comprehensive analysis and understanding of the challenges in simultaneous speech translation.

**Reasons To Reject:**

The use of external alignment tools and pretrained models like wav2vec2.0 could potential make the method less effective in other scenarios with low-resource language pairs and non-English source. But I think that given the benefits and the known difficulty of the task, it is not a compelling reason of concern.

**Reproducibility:**

4: Could mostly reproduce the results, but there may be some variation because of sample variance or minor variations in their interpretation of the protocol or method.

**Reviewer Confidence:**

4: Quite sure. I tried to check the important points carefully. It's unlikely, though conceivable, that I missed something that should affect my ratings.

**Typos Grammar Style And Presentation Improvements:**

Figure 2: consider using a variable (n) instead of an actual number (3), for the number of Transformer Layers.

---

> ### Author Rebuttal · Authors · 2023-08-29
>
> #### Thank you for your valuable feedback and suggestions. We’ll go over your concerns below.
>
>
> > Q1: What version of wav2vec 2.0 are you using? Is the non-ASR fine-tuned base model?
>
> A1: Yes, thank you for your domain expertise and familiarity with experimental details. We indeed use the non-ASR fine-tuned base model of wav2vec 2.0.
>
>
> > Q2: Did you carry out any ablation studies regarding the impact of mixup (Fang et al 2022) during the pre-training?
>
> A2: We did not perform ablation studies specifically on the impact of mixup. We appreciate your professional suggestion and have included additional experiments to address this.
>
> | Mixup | Accuracy | WER | Avg length bias (L2) |
> | --- | :---: | :---: | :---: |
> | + | 56.83 | 17.41 | 2.099 |
> |    | 56.97 | 17.66 | 2.110 |
>
>
> We are somewhat surprised as well. It appears that mixup has minimal impact on our pre-training.
> We will further clarify the scope of our research.
>
> > Q3: In lines 410-411, it is stated that no additional ASR data are used. However, doesn't the use of the external alignment tool indirectly imply the use of ASR data?
>
> A3: Thank you for raising this insightful question. You are correct that the use of the external alignment tool, MFA, implies the use of Librispeech ASR data. We utilized MFA as a readily available alignment tool instead of training our own alignment tool.
>
> In theory, training our own MFA alignment model with Must-C ASR data might yield better results. Here are a couple of reasons for considering this:
>
> (1) There is a domain gap between Librispeech and Must-C, and our previous experience suggests that incorporating more Must-C ASR data yields better results for English ASR tasks. For example, in a previous work using an Encoder-Decoder architecture for En-De Must-C ST data, the word error rate (WER) was 18.2. After incorporating Librispeech, the WER decreased to 17.8. However, using all eight language pairs of Must-C ASR data reduced the WER to 16.8, even though some ASR data for other language pairs overlapped with the En-De ASR data. This trend was also observed in the IWSLT competition, where training with a larger dataset like Must-C V2 yielded better results than Must-C V1.
>
> (2) MFA might not provide perfect alignment. On one hand, MFA can not handle noisy or speech signals that cannot extract CMVN features, resulting in the exclusion of some training data (after filtering, train data size is reduced from 225K to 217K). Following the approach of SpeechUT, training an encoder-decoder model for speech-to-text unit alignment using a small portion of ASR data could be beneficial. In theory, we could leverage this model to train our own speech-text alignment tool, which would prevent us from losing a portion of the training data.
>
> Therefore, training an in-domain MFA model using Must-C data might lead to better results. We acknowledge the value of your professional suggestion and will continue exploring these possibilities in the future.
>
> Thank you for your insightful input. This is indeed an area worth further investigation.
>
> And we'll revise the next edition in writing according to your suggestions.

---

### Official Review · Reviewer_j7oz · 2023-08-04

**Soundness:** 3

**Excitement:**

3: Ambivalent: It has merits (e.g., it reports state-of-the-art results, the idea is nice), but there are key weaknesses (e.g., it describes incremental work), and it can significantly benefit from another round of revision. However, I won't object to accepting it if my co-reviewers champion it.

**Paper Topic And Main Contributions:**


This paper introduces a two-stage training method for simultaneous speech translation. It first conducts a token-level semantic alignment pre-training between audio and transcription. Then it utilizes a robust and random wait-k tokens policy to enable simultaneous decoding.
The authors verify the proposed method on MuST-C English to German, French, and Spanish speech translation pairs. Results demonstrate the effectiveness of the method.


**Questions For The Authors:**


See above.

**Reasons To Accept:**



1. This paper introduces a cross-modal pre-training method for acoustic model, with the fine-grained CIF module. And a robust wait-k tokens strategy is proposed to optimize the simultaneous speech translation tasks.

2. The proposed method achieves a better trade-off between translation quality and latency than baselines.

**Reasons To Reject:**



1. The training processing of the proposed method is more complicated. It has multiple training stages and losses.

2. Why is your model significantly better than other models in Figure3? This question needs to be explained clearly.

**Reproducibility:**

2: Would be hard pressed to reproduce the results. The contribution depends on data that are simply not available outside the author's institution or consortium; not enough details are provided.

**Reviewer Confidence:**

4: Quite sure. I tried to check the important points carefully. It's unlikely, though conceivable, that I missed something that should affect my ratings.

---

> ### Author Rebuttal · Authors · 2023-08-29
>
> #### Thank you for your helpful reviews and comments. We will address your concerns individually.
>
> > Q1: The training processing of the proposed method is more complicated. It has multiple training stages and losses.
>
> A1: Thanks for proposing this concern. Similar to most speech translation works (e.g., STPT, SpeechUT, MoSST), we have only two standard training stages: pre-training and fine-tuning. Some work (e.g., MAESTRO) even requires three stages.
>
> Our approach is a simultaneous ST model for streaming input.
>
> - In pre-training stage, our main objective is speech-text alignment for streaming input. We have `3` losses, where $L_{ce}$ and $L_{emb}$ aim at speech-text alignment, and fine-grained $L_{cif}$ aim at audio segmentation for streaming input.
> - In fine-tuning stage, our main objective is simultaneous translation with `2` additional losses -- offline and streaming translation losses, where they are applied to two modalities: speech and text, i.e., $L_{off}^{st}, L_{off}^{mt}, L_{wk}^{st}, L_{wk}^{mt}$
>
> For this response, we compare with a non-streaming ST model - STPT, which has `4` losses and `1` additional model.
> - Their masked prediction loss $L_{ssl}$ and the speech phoneme classification loss $L_{s2p}$ are used for speech-text alignment. The two losses are equivalent to our $L_{ce}$ and $L_{emb}$
> - Their $L_{s2p}$ loss also requires to train an additional `HMM-GMM model` for label annotations.
> - Their text-to-text loss $L_{t2t}$ and speech-to-text loss $L_{s2t}$ are exactly our $L_{off}^{mt}, L_{off}^{st}$.
>
> In contrast, our fine-grained $L_{cif}$ and $L_{wk}^{st}, L_{wk}^{mt}$ are newly introduced, but they are both designed for simultaneous scenario.
>
> [1] STPT:  Yun Tang, Hongyu Gong, Ning Dong, Changhan Wang, Wei-Ning Hsu, Jiatao Gu, Alexei Baevski, Xian Li, Abdelrahman Mohamed, Michael Auli, and Juan Pino. 2022. Unified speech-text pre-training for speech translation and recognition. In Proceedings 874
> of the 60th Annual Meeting of the Association for Computational Linguistics (ACL Volume 1: Long Papers), pages 1488–1499, Dublin, Ireland. Association for Computational Linguistics.
>
> [2] SpeechUT:  Ziqiang Zhang, Long Zhou, Junyi Ao, Shujie Liu, Lirong Dai, Jinyu Li, and Furu Wei. 2022. SpeechUT: Bridging Speech and Text with Hidden-Unit for Encoder-Decoder Based Speech-Text Pre-training. In Proceedings of the 2022 Conference on Empirical Methods in Natural Language Processing (EMNLP), pages 1663–1676, Abu Dhabi, United Arab Emirates. Association for Computational Linguistics.
>
> [3] MoSST:   Qian Dong, Yaoming Zhu, Mingxuan Wang, and Lei Li. 2022. Learning when to translate for streaming speech. In Proceedings of the 60th Annual Meeting of the Association for Computational Linguistics (ACL Volume 1: Long Papers), pages 680–694, Dublin, Ireland. Association for Computational Linguistics.
>
> [4] MAESTRO:  Zhehuai Chen and Yu Zhang, Andrew Rosenberg, Bhuvana Ramabhadran, Pedro J. Moreno, Ankur Bapna, and Heiga Zen. MAESTRO: Matched Speech Text Representations through Modality Matching. 2022 Interspeech.
>
>
> > Q2: Why is your model significantly better than other models in Figure3? This question needs to be explained clearly.
>
> A2: Thanks for raising this issue. We've already mentioned two main reasons in our paper.
> 1. In the abstract and the paragraph in L80 of introduction, we mentioned that our approach finds an appropriate method for bridging the gap between streaming MT and streaming ST. The fine-grained CIF module successfully transforms the audio feature from audio length into transcription length. Then we can leverage the text simultaneous translation tricks to improve simultaneous speech translation.
> 2. In L419 and L431 of experimental section, we mentioned that En-De is explored by many related SimulST works while En-Fr and En-Es are seldom researched. Therefore, our improvement on En-De over other baselines is not as significant as the other two language pairs.
>
> A possible reason that is not discussed in the paper is that our method combines the advantages of previous offline translation models (e.g., forced alignment trick) and simultaneous speech translation models (e.g., CIF audio segmentation trick). Our pre-training with MFA annotation and fine-tuning with random/robust wait-k-tokens strategy also make the transition from offline ST (infinity latency) to SimulST (small and finite latency) much smoother. Therefore, we maintain good ST performance for low latency region.
>
> ### Reproducibility: 2 The contribution depends on data that are simply not available outside the author's institution or consortium; not enough details are provided.
> We have provided all the hyper-parameter settings, training steps, and model architecture sizes in the paper. Additionally, we have uploaded the code package, which includes script files for each step of the process. The data we used is open-sourced as well and underwent the same data preprocessing steps as STEMM.

---

### Official Review · Reviewer_7R6j · 2023-08-11

**Soundness:** 4

**Excitement:**

4: Strong: This paper deepens the understanding of some phenomenon or lowers the barriers to an existing research direction.

**Missing References:**

In addition to the references cited in "Reasons to reject", I think the paper should include the following additional references:
- L133: Le et al. 2023 (Pre-training for Speech Translation: CTC Meets Optimal Transport) performs a comprehensive study on the effects of supervised pre-training to ST training. It specifically aims to better align speech and text at the pre-training stage, which is highly related to this work. It also demonstrates the benefits of using CTC loss in pre-training (which is relevant to Table 4 of this work).
- L137: important papers in multi-task learning such as Bahar et al. 2019 (A Comparative Study on End-To-End Speech to Text Translation) should be mentioned as well.
- L139: First work which adopts knowledge distillation for ST such as Liu et al., 2019 (End-to-End Speech Translation with Knowledge Distillation) should be appreciated.
- As a minor suggestion, a wait-$k$ strategy with fixed window is also proposed for offline ST in Le et al. 2020 (Dual-decoder Transformer for Joint Automatic Speech Recognition and Multilingual Speech Translation).
- Table 2 should include reference to the current SoTA results in ST of SpeechUT (Zhang et al. 2022 "SpeechUT: Bridging speech and text with hidden unit for encoder-decoder based speech-text pre-training") and CRESS (Fang et Feng. 2023 "Understanding 668 and bridging the modality gap for speech translation").

**Paper Topic And Main Contributions:**

This paper proposes pre-training and fine-tuning methods for simultaneous speech-to-text translation (ST). The pre-training step which focuses on aligning speech and text sequences in terms of both length and representation space is based on the Continuous Integrate-and-Fire (CIF) module. Additional pre-training losses including L2-distance and cross-entropy (CE) loss are also used. The fine-tuning phase is performed in a multi-task learning setup with multiple objectives, including the ones in the pre-training stage. A novel wait-$k$ strategy is proposed for the fine-tuning stage to improve robustness of the model in various latency settings.

**Questions For The Authors:**

- L211-216: could you please clarify this point? It is not clear to me why the cropping of 320k audio samples produces the alignment between acoustic-to-tokens $a$ and the word-level transcription.
- Eq2: does it mean that $b_1 = 1$ and $e_w = T$?
- L239: Is this the quantity loss proposed in Dong and Xu, 2020?
- L271:  $z = Linear_{emb}(h)$: Is this linear layer initialized from pre-trained NMT model and frozen during training?

**Reasons To Accept:**

- The paper is well-written with clear motivation and easy-to-follow structure.
- The proposed method has the potential to be applied in various speech applications other than simultaneous and offline ST as speech-text alignment is an important problem in many speech-related tasks.
- The ablation study and analysis is interesting, offering insights into the effectiveness of the proposed components in different settings (online and offline).
- The method performs well on both simultaneous and offline ST tasks and seems to be robust for different latency requirements.

**Reasons To Reject:**

1) Related work section misses a highly relevant line of work that aims to align speech and text in speech in general (and in ST in particular) that would help to better position itself in the field.

	- Different alignment methods have been proposed for self-supervised pre-training for speech such as Bapna et al. 2021 (SLAM: A unified encoder for speech and language modeling via speech-text joint pre-training), Chen et al. 2022 (MAESTRO: matched speech text representations through modality matching), Bapna et al. 2022 (mSLAM: Massively multilingual joint pre-training for speech and text), for supervised ST pre-training such as Alinejad and Sakar. 2020 (Effectively pretraining a speech translation decoder with machine translation data) and Le et al. 2023 (Pre-training for Speech Translation: CTC Meets Optimal Transport), or in a multi-task learning framework for ST such as Liu et al. 2020 (Bridging the modality gap for speech-to-text translation), Tang et al. 2021 (Improving speech translation by understanding and learning from the auxiliary text translation task), Ye et al. 2022 (Cross-modal contrastive learning for speech translation), Ouyang et al. 2022 (WACO: word-aligned contrastive learning for speech translation), etc., just to name a few off the top of my head.

2) The analysis of the speech-text alignment method in the pre-training stage is not very convincing to me. In particular,
	- The contribution of each auxiliary loss in the pre-training stage to the final ST performance is not analyzed.
	- The proposed pre-training method should be compared against other pre-training methods (at least the most common method).

3) The fine-tuning stage consists of many objectives (7 individual losses) and the paper lacks an analysis of the contributions of these individual losses to the final ST performance. For example, whether $L_{pt}$ is really needed during the fine-tuning stage?

**Reproducibility:**

4: Could mostly reproduce the results, but there may be some variation because of sample variance or minor variations in their interpretation of the protocol or method.

**Reviewer Confidence:**

4: Quite sure. I tried to check the important points carefully. It's unlikely, though conceivable, that I missed something that should affect my ratings.

**Typos Grammar Style And Presentation Improvements:**

- It is not very clear from Figure 2 what the role of NMT modules are.
- Interesting visualization such as Figure 8 should be included in the main content.
- Figure 6: Should add caption referring to Table 3 so that readers know what models a-g mean.

---

> ### Author Rebuttal · Authors · 2023-08-29
>
> #### Thank you for your positive reviews and your familiarity with the relevant works in our field. We will address your concerns as follows.
>
> >Q1: The related work section is missing some relevant citations.
>
> A1: We appreciate your attention to our field and expertise. We will add these citations in the revised version. We will also emphasize that our pre-training method mainly focuses on streaming scenario with fine-grained CIF module, while other works (e.g., MAESTRO, mSLAM, etc.) focus on offline scenario. In addition, the two different objectives may have different data requirements, e.g., regular speech pre-training may only need audio-only data without transcription.
>
> Compared with suggested speech translation related works, our work mainly focuses on the simultaneous speech translation task with ST and MT data, while other works explore better speech representation with extensive unlabeled audio. Therefore, in our current version, we did not conduct a comparison with them.
> For example,  Ye et al. 2022 in the suggested references has been already compared (as ConST) in our Table 2.
>
> >Q2: Insufficient analysis of the speech-text alignment method in the pre-training stage.
> - Q2-a: The contribution of each auxiliary loss in the pre-training stage to the final ST performance is not analyzed.
>
> A2-a: The three pre-training losses (CE loss, embed loss, and fine-grained CIF loss) are by default optimized in fine-tuning stage, and the relevant ablation study is mainly conducted on the fine-tuning stage in Table 3.
>
> For this response, we conduct several new ablation studies of the auxiliary loss in the pre-training stage. In the following Table, if the loss is not marked as `+`, it means the corresponding loss is removed in both pre-training and fine-tuning stages.
>
> | CE loss | embed loss | fine-grained CIF | BLEU（beam5） |
> | :---: | :---: | :---: |:---: |
> | + | + | + | 29.15 |
> | + |    | + | 28.77 |
> |    |    | + | 27.84 |
>
> According to the above Table, we can observe that removing only the embed loss has little impact on the final results, but we also notice that it will significantly affect the convergence speed during fine-tuning. Normally, the best dev BLEU is achieved around the 10th epoch, but without the embed loss, it becomes the 13th epoch.
>
> Removing both the CE loss and embed loss in both the pre-training and fine-tuning stages results in a significant drop in performance. Based on the previous ablation experiment results, we can conclude that the CE loss plays a crucial role in the pre-training stage but has minimal impact in the fine-tuning stage.
>
> - Q2-b: The proposed pre-training method should be compared against other pre-training methods
>
> A2-b: Thanks for the insightful suggestion. In general, the pre-training model is difficult to compare directly, and it requires some downstream tasks for evaluation. Therefore, we compared with two pre-training based SimulST models (MoSST and MU-ST) in Figure 2.
>
> In addition, our pre-training is designed for streaming scenario with the proposed fine-grained CIF module, while other works focus on offline pre-training. It is also unfair for direct comparison. Since our model is comparable with offline ST, we compared with STPT in Table 2, where STPT employs an offline (token masking with bidirectional attention) pre-training framework. Note that STPT used the comparable amount of ASR data but extremely large-scale unlabeled audio (60K hours) in pre-training.
>
> >Q3: The fine-tuning stage lacks sufficient ablation experiments. For example, is $L_{pt}$ really needed during the fine-tuning stage?
>
> A3: Whether applying pre-training stage has been discussed in Table 3, and we also include some ablation studies on individual pre-training loss in Table 3. We would like to conduct more experiments to address the raised concern.
>
> | $L_{pt}$ (when finetune) | BLEU（beam5） | BLEU（beam1） |
> | --- | :---: | :---: |
> | + | 29.15 | 28.12 |
> |    | 24.49 | 23.32 |
>
> In the bottom row of above Table, after the pre-training stage, all three losses in $L_{pt}$ are removed in fine-tuning. As expected, the BLEU on the dev set significantly dropped.
>
> Theoretically, the fine-grained CIF loss in $L_{pt}$ is crucial because it provides the ability of audio segmentation for the streaming input.
> Without CIF loss, we found the length prediction becomes extremely worse, and impose a big gap between the training and inference. In addition, the convergence becomes disordered. The convergence is usually achieved after 10+ epochs. Surprisingly, when $L_{pt}$ is removed, the highest BLEU on dev set is observed in the first epoch, followed by a decline in subsequent epochs. Therefore, our results indicate that $L_{pt}$ is indeed crucial during the fine-tuning stage. Actually, including the $L_{pt}$ in fine-tuning stage becomes common in speech translation, e.g., offline model STPT and Simul model MoSST.
>
> ### Questions for the authors:
>
> >Q1: L211-216: Could you please clarify this point? It is not clear to me why the cropping of 320k audio samples produces the alignment between acoustic-to-tokens $a$ and the word-level transcription.
>
> A: There may be some misleading in this sentence. We did not apply `cropping` of `320k` audio samples. We did a `downsampling` by `320x` on the audio frames, i.e., reducing the audio length by 320x.
>
> In the previous sentence, we mentioned that we used the MFA tool to achieve word-level alignment between transcripts and raw audio frames. Due to the 320x compression through the 7-layer CNN downsampling of wav2vec2, the alignment after compression can still be achieved because the strides of CNN layers are known.
>
> >Q2: Does Eq2 mean that $b_1 = 1$ and $e_w = T$?
>
> A: Yes, that is correct, $b_1 = 1$ and $e_w = T$.
>
> >Q3: Is the quantity loss proposed in Dong and Xu, 2020 referenced in L239?
>
> A: Yes, the quantity loss mentioned in line 239 is proposed in Dong and Xu, 2020, which is the traditional CIF loss. We proposed the fine-grained CIF loss.
>
> >Q4: L217:$z = Linear_{emb}(h)$ Is this linear layer initialized from pre-trained NMT model and frozen during training?
>
> A: The linear layer maps the speech representation $h$ to the source embedding space, $R^{d} \rightarrow R^{d}$, where $d$ is hidden dimension (equal to embedding dimension in our case). This linear layer did not exist in the NMT model, and it is trained in both stages.
>
>
> ### Missing References:
>
> We appreciate your professional suggestions and will add relevant references in the revised version.
>
> We will also mention that the suggested references are all non-streaming ST models, which are difficult to compare with our proposed simultaneous approach. For example, it is unfair to plot the common BLEU-AL curve for non-streaming ST models.
>
> Since our model is comparable with non-streaming ST task, we would like to include the recent offline ST works. E.g., Le et al. 2023 was released in May, 2023 and we did not notice this paper when preparing our submission. CRESS has two versions by being built upon HuBERT and Wav2Vec2. Our model is built upon Wav2Vec2 and achieves a better result than CRESS Wav2Vec2, but worse than CRESS HuBERT. We will also add the SOTA of non-streaming ST model - SpeechUT in the revised version.

---

### Meta-Review · Area_Chair_wVAq · 2023-09-14

**Recommendation:** 5

**Metareview:**

The paper proposes a new approach to simultaneous speech translation, where real-time translations are produced by a system as audio is incrementally consumed. Reviewers were impressed with the proposed method, both in terms of soundness and novelty, especially after the authors addressed some of the concerns during the rebuttal period. The reviewers also commented positively on the paper's writing and the thoroughness of the experiments. The result is a good piece of work that could have wide interest.

---

### Decision · Program_Chairs · 2023-10-07

**Decision:**

Accept-Main

**Comment:**

The paper proposes a new approach to simultaneous speech translation, where real-time translations are produced by a system as audio is incrementally consumed. Reviewers were impressed with the proposed method, both in terms of soundness and novelty, especially after the authors addressed some of the concerns during the rebuttal period. The reviewers also commented positively on the paper's writing and the thoroughness of the experiments. The result is a good piece of work that could have wide interest.